# Umbilical Cord Plasma Lysophospholipids and Triacylglycerols Associated with Birthweight Percentiles

**DOI:** 10.3390/nu16020274

**Published:** 2024-01-17

**Authors:** Gerard Wong, Kothandaraman Narasimhan, Wei Fun Cheong, Sharon Ng, Izzuddin M. Aris, See Ling Loy, Anne K. Bendt, Kok Hian Tan, Fabian K. P. Yap, Lynette P. Shek, Yap Seng Chong, Peter D. Gluckman, Keith M. Godfrey, Yung Seng Lee, Markus R. Wenk, Neerja Karnani, Shiao-Yng Chan

**Affiliations:** 1Singapore Institute for Clinical Sciences (SICS), A*STAR, Singapore 117609, Singaporekothandaraman_narasimhan@sics.a-star.edu.sg (K.N.); pd.gluckman@auckland.ac.nz (P.D.G.); neerja_karnani@sics.a-star.edu.sg (N.K.); 2Singapore Lipidomics Incubator, Department of Biochemistry, Yong Loo Lin School of Medicine, National University of Singapore, Singapore 117593, Singapore; obgcwf@nus.edu.sg (W.F.C.); anne.bendt@nus.edu.sg (A.K.B.); bchmrw@nus.edu.sg (M.R.W.); 3Department of Obstetrics and Gynaecology, Yong Loo Lin School of Medicine, National University of Singapore (NUS), Singapore 117593, Singapore; sharon_ng@sics.a-star.edu.sg (S.N.); obgcys@nus.edu.sg (Y.S.C.); 4Division of Chronic Disease Research across the Lifecourse, Department of Population Medicine, Harvard Medical School and Harvard Pilgrim Health Care Institute, Boston, MA 02215, USA; izzuddin_aris@hphci.harvard.edu; 5KK Women’s and Children’s Hospital (KKH), Singapore 229899, Singapore; loyseeling@duke-nus.edu.sg (S.L.L.); tan.kok.hian@singhealth.com.sg (K.H.T.); fabian.yap.k.p@singhealth.com.sg (F.K.P.Y.); 6Department of Pediatrics, Yong Loo Lin School of Medicine, National University of Singapore, Singapore 119228, Singapore; paeshekl@nus.edu.sg (L.P.S.); paeleeys@nus.edu.sg (Y.S.L.); 7MRC Lifecourse Epidemiology Centre, University of Southampton, University Hospital Southampton NHS Foundation Trust, Southampton SO16 6YD, UK; 8NIHR Southampton Biomedical Research Centre, University of Southampton, University Hospital Southampton NHS Foundation Trust, Southampton SO16 6YD, UK

**Keywords:** umbilical cord blood, lysophosphatidylcholine, triacylglycerol, lipidomics, fetal growth, GUSTO

## Abstract

Dysregulated transplacental lipid transfer and fetal–placental lipid metabolism affect birthweight, as does maternal hyperglycemia. As the mechanisms are unclear, we aimed to identify the lipids in umbilical cord plasma that were most associated with birthweight. Seventy-five Chinese women with singleton pregnancies recruited into the GUSTO mother–offspring cohort were selected from across the glycemic range based on a mid-gestation 75 g oral glucose tolerance test, excluding pre-existing diabetes. Cord plasma samples collected at term delivery were analyzed using targeted liquid-chromatography tandem mass-spectrometry to determine the concentrations of 404 lipid species across 17 lipid classes. The birthweights were standardized for sex and gestational age by local references, and regression analyses were adjusted for the maternal age, BMI, parity, mode of delivery, insulin treatment, and fasting/2 h glucose, with a false discovery-corrected *p* < 0.05 considered significant. Ten lysophosphatidylcholines (LPCs) and two lysophosphatidylethanolamines were positively associated with the birthweight percentiles, while twenty-four triacylglycerols were negatively associated with the birthweight percentiles. The topmost associated lipid was LPC 20:2 [21.28 (95%CI 12.70, 29.87) percentile increase in the standardized birthweight with each SD-unit increase in log_10_-transformed concentration]. Within these same regression models, maternal glycemia did not significantly associate with the birthweight percentiles. Specific fetal circulating lysophospholipids and triacylglycerols associate with birthweight independently of maternal glycemia, but a causal relationship remains to be established.

## 1. Introduction

### 1.1. Maternal Glycemia and Offspring Birthweight

Maternal glycemia has been strongly correlated with offspring birthweight as a continuum across the glycemia spectrum [1]. Further, there have been reports showing associations between well-controlled gestational diabetes (GDM) and fetal overgrowth, or macrosomia, indicating that factors other than glucose may actively contribute to fetal growth [2,3]. For example, in pregnant women with well-controlled GDM, maternal circulating free fatty acids were positively correlated with neonatal adiposity at birth [4]. This observation has led to the hypothesis that dysregulated maternal-to-fetal lipid transfer and placental lipid metabolism may also be critical drivers in the regulation of fetal size, independently of maternal glycemia [5]. However, the mechanisms remain poorly understood.

### 1.2. Placental Lipid Metabolism and Transplacental Lipid Supply

The transplacental supply of glucose and key amino acids are known to influence fetal growth but the specific lipid species involved remain unknown. Lipids not only form components of cellular structures and serve as substrates for energy and metabolism, but they have wide-ranging bioactive properties of their own [6]. It is believed that fatty acids are transferred in unesterified forms across the maternal–placental and placental–fetal interfaces, but there is now evidence that lipids such as lysophospholipids may also cross the placental syncytiotrophoblast barrier through plasma membrane transporters [7]. It is clear that fatty acids and lipids derived from the mother undergo extensive metabolism in the placenta before export to the fetus [8,9] for nutritive purposes and also as biological signaling compounds. In addition to transplacental transfer, fetal fatty acids and lipids can also be synthesized from glucose by fetal tissues de novo, except the long-chain poly-unsaturated fatty acids (LC-PUFAs) due to the lack of specific desaturase expression in fetal–placental tissue. However, the full range of fatty acids and the predominant lipid forms in which fatty acids are transported across the placental interfaces are not well-characterized.

An understanding of the key lipid species involved in regulating fetal size and the role of placental lipid metabolism and transfer are essential in forming the basis for developing novel approaches in the management of pathological conditions such as gestational diabetes, maternal obesity, and uteroplacental insufficiency. New strategies, including the modulation of placental lipid metabolism and transplacental lipid supply, are required since the present approaches aimed at improving maternal glycemic regulation and the control of gestational weight gain have not shown consistent efficacy in optimizing fetal growth.

### 1.3. Hypothesis and Aims

We hypothesized that specific lipid species in the fetal circulation are associated with fetal growth. With increasing maternal glycemia, we postulate that there is altered transplacental lipid supply and changes in the fetal metabolism of lipids, which collectively alter the concentration of specific lipid species in the fetal circulation that are associated with fetal size.

To address the present knowledge gaps, we aimed to describe the relationship between maternal glycemia and the concentrations of specific lipid species in umbilical cord plasma, identify the lipid species in umbilical cord plasma which are most associated with birthweight, and assess, alongside each specific lipid species, the estimated effect of maternal glycemia on birthweight.

## 2. Materials and Methods

### 2.1. The GUSTO Cohort and Sample Selection for This Sub-Study

This work is part of The Growing Up in Singapore Towards Healthy Outcomes (GUSTO) study, a prospective mother–offspring cohort study in Singapore designed to investigate the effect of early life events on the risk of developing metabolic diseases later in life [10]. This study received ethical approval from the Centralized Institutional Review Board of the KK Women’s and Children’s Hospital (KKH) and the Domain-Specific Review Board of the National University Hospital (NUH). Informed written consent was obtained from all participants upon recruitment. Pregnant women (*n* = 1450) were recruited at 10 to 14 weeks of gestation between June 2009 and September 2010 from the two largest public maternity units in Singapore, KKH and NUH. Eligible participants were Singapore residents of Chinese, Malay, or Indian ethnicity, willing to donate bio-samples. Further details of the overall GUSTO study have been described elsewhere [10]. The majority of participants completed a two time-point 75 g oral glucose tolerance test (OGTT; *n* = 1136) and provided cord blood samples (*n* = 886).

For this sub-study, cases were selected based on availability of cord plasma samples and data on OGTT, pre-pregnancy BMI and ethnicity. Known cases of pre-existing maternal diabetes, multiple gestation, and preterm delivery, were excluded. To facilitate further selection, both the fasting and 2 h glycemia values were each considered in quartiles (as calculated across the whole cohort); cases where both the fasting and 2 h glycemia lay in the same quartile were considered for selection. Cases were then also confined to those of neonates with homogenous Chinese parentage (the largest ethnic group in GUSTO) [9] and delivered by non-smoking mothers, to minimize confounding by these factors. This resulted in a final sample of 75 cases (study selection flow chart in Appendix A), which incidentally comprised a similar number of cases in each of the four glycemia quartiles: 19, 21, 17, and 18 cases in quartiles one to four, respectively.

### 2.2. Data and Cord Blood Collection

Demographic data including maternal age, ethnicity, self-reported pre-pregnancy body mass index (BMI; which was highly correlated with measured first trimester BMI in the overall GUSTO cohort, correlation coefficient 0.96, *p* < 0.001), and history of assisted conception were obtained by interviewer-administered questionnaires during pregnancy, and details of parity, mode of delivery, infant sex, gestational age at birth, and birthweight were obtained from medical records.

Umbilical venous cord blood was collected at delivery in EDTA tubes, centrifuged and the plasma stored frozen with trasylol until batch analysis.

### 2.3. Cord Blood Plasma Lipidomic Analysis

Thawed cord plasma samples were spiked with known amounts of lipid internal standards and lipids were extracted using a modified Bligh and Dyer method [11]. The lipidomic profile of the cord plasma samples were determined with targeted lipid analysis using liquid chromatography multiple reaction monitoring mass spectrometry (LC-MS/MS). The quantification of phospholipids, sphingolipids, and glycolipids were determined by an Agilent 1290 HPLC system (Agilent Technologies Inc., Waldbronn, Germany) paired with an Agilent 6460 triple quadrupole mass spectrometer (Agilent Technologies Inc., Santa Clara, CA, USA) [12]. The quantification of neutral lipid species diacylglycerols (DAGs) and triacylglycerols (TAGs) was determined by a Shimadzu Prominence Ultra Fast Liquid Chromatography (UFLC) system (Shimadzu Corporation, Duisburg, Germany), paired with an ABI 3200 triple quadrupole mass spectrometer (Sciex, Framingham, MA, USA). Samples were analyzed in a single batch where quality control (QC) samples, prepared by pooling lipid extracts from all samples, were inserted after every 10 samples. Signals falling below the limit of quantitation (LOQ) were assigned a value of 0. In total, the concentrations of 404 lipid species from 17 lipid classes were determined; peak area for each species was normalized against respective internal standards. Concentrations of these lipids are detailed in Appendix A. The 17 lipid classes included TAG, DAG, lysophosphatidylcholine (LPC), lysophosphatidylethanolamine (LPE), ceramide (Cer), dihydroceramide (DHCer), monohexosyl ceramide (MHCer), phosphatidylcholine (PC), oddPC, alkyl-phosphatidylcholine [PC(O)], alkenyl-phosphatidylcholine [PC(P)], phosphatidylethanolamine (PE), alkyl-PE [PE(O)], alkenyl-PE [PE(P)], phosphatidylinositol (PI), phosphatidylserine (PS), and sphingomyelin (SM).

### 2.4. Statistical Analysis

The sex-specific birthweight-for-gestational age percentile was derived based on GUSTO data according to published methods [13]. The concentrations of lipid species were log_10_ transformed and standardized to z-scores to minimize skewness in their distribution and to facilitate comparison between lipid species with often very different levels of concentration. Multiple linear regression was applied for each and every lipid species analyzed in cord blood plasma to determine its association with maternal glycemia or standardized birthweight percentile. Adjustments were made for maternal age, maternal BMI, parity, mode of delivery (vaginal, non-labor cesarean, intrapartum cesarean), conception by assisted reproductive techniques, insulin treatment, fasting glucose, and/or 2 h post-load glucose. Fasting and 2 h post-load glucose were checked for multicollinearity to prevent over-adjustment within the same model. Gestational Diabetes Mellitus (GDM) was defined using the World Health Organization (WHO) 1999 criteria (fasting glucose ≥ 7.0 and/or 2 h glucose ≥ 7.8 mmol/L), which were the criteria in use at the time of the study. Logistic regression adjusting for the same covariates were additionally performed to associate lipids with the mutually exclusive categories of small-for-gestational-age (SGA; birthweight <10th percentile) and large-for-gestational-age (LGA; >90th percentile), each group compared with the appropriately-grown-for-gestational age (AGA; 10–90th percentile) group. The *p*-values were false-discovery rate (FDR) corrected using the Benjamini–Hochberg approach [14]. Statistical significance was set at FDR < 0.05.

We also conducted a series of pre-planned sensitivity analyses to assess the robustness of our results. Three sub-analyses were performed to ensure results were not significantly changed: (i) exclusion of those treated with insulin, (ii) exclusion of those with diabetes in pregnancy as determined by fasting glucose ≥ 7.0 and/or 2 h glucose ≥ 11.1 mmol/L, (iii) exclusion of possible placental insufficiency suggested by a birthweight percentile below 10, or the presence of chronic hypertension, pre-eclampsia or pregnancy-induced hypertension. All statistical analyses were performed using MATLAB R2017b or R v4.3.1 (Vienna, Austria).

## 3. Results

### 3.1. Cohort Characteristics

Among the 75 cases included in this sub-study, the mean fasting and 2 h post-load glucose were 4.42 and 6.60 mmol/L, respectively. (Table 1). Fifteen women (20%) had GDM and three of them received insulin treatment. Most of the women had a normal pre-pregnancy BMI with a cohort mean of 21.78 kg/m^2^. The mean gestational age at birth was 39.2 weeks, and the average standardized birthweight percentile was 53.55%; 50.7% were female infants with 68% were delivered vaginally. A small minority (8%) were conceived by assisted conception.

### 3.2. Association of Cord Blood Lipid Species with Maternal Glucose

Although 78 lipid species demonstrated an uncorrected *p*-value < 0.05 (filtered results shown in Table 2), after correcting for false discovery, none of the 404 cord blood lipid species’ concentrations were associated with fasting glucose after adjustment for the maternal age, maternal BMI, mode of delivery, assisted conception, and insulin treatment.

The filtered (uncorrected *p* < 0.05) results for the association of the cord blood lipid species concentration with the 2 h post-load glucose adjusted for similar covariates as for the fasting glucose, included 83 lipid species (Table 3). Following FDR correction, a higher 2 h-glucose was significantly associated with only one lipid species: ceramide (Cer) d18:1/C33:1 (β 0.36 SDs of log_10_ concentration per mmol/L increase; 95% CI 0.21, 0.51; FDR corrected *p* = 4.93 × 10^−3^).

### 3.3. Association of Cord Blood Lipid Species with Birthweight

After adjusting for the maternal age, pre-pregnancy BMI, parity, mode of delivery, assisted conception, fasting glucose, 2 h glucose, and insulin treatment, only 36 lipid species from three lipid classes were identified to be significantly associated (FDR-corrected *p*-value < 0.05) with the standardized birthweight percentiles: specifically, lysophosphatidylcholine (LPC; 10 species), lysophosphatidylethanolamines (LPE; 2 species), and triacylglycerols (TAG; 24 species) (Table 4). The LPC and LPE lipid species were all positively associated, while the TAGs were all negatively associated, with the standardized birthweight percentiles. 

The LPC species containing eicosadienoic acid (LPC 20:2) displayed the largest magnitude of association and most statistically significant result with the standardized birthweight percentiles. Each SD-unit increase in the log_10_-transformed concentration of LPC 20:2 was associated with a 21.28 (95%CI 12.70, 29.87) percentile increase in the standardized birthweight. Lysophospholipids containing mono- and poly-unsaturated fatty acids of 18-carbon and 20-carbon chain lengths also featured prominently: LPC 18:1, LPC 18:2, LPC 18:3, LPC 20:0p/20:1e, LPC 20:1, LPC 20:2, LPC 20:3, and LPC 20:4. The lysophospholipids containing the long-chain poly-unsaturated fatty acids (LC-PUFA) arachidonic acid (LPC 20:4), docosapentaenoic acid (DPA; LPC 22:5), as well as containing a common mono-unsaturated fatty acid (LPC 16:1) were also identified. Of note, none of the identified LPCs or LPEs contained saturated fatty acids.

Generally, the TAGs displayed a similar magnitude of association with the standardized birthweight percentiles (range: −12.11 to −27.67% per-SD increase log_10_-transformed lipid concentration) compared with the lysophospholipids (13.19 to 21.28% per-SD log_10_ lipid), but the associations were in the opposite direction (Table 4). The lipid species with the strongest negative association was TAG 50:2 (β −27.67% [−41.81, −13.52] per-SD log_10_ lipid). The fatty acid triplet composition of many of these TAGs (total carbon ranged from 48–58) comprised predominantly long-chain fatty acids (≥14 carbon length), and 83.3% of them contained at least one mono-unsaturated or poly-unsaturated fatty acid combined with one or two long-chain saturated fatty acids. Interestingly, 20.8% of the TAGs (TAG 53:1, TAG 55:3, TAG 57:3, TAG 53:3, TAG 53:0) included at least one odd chain fatty acid.

Within the same linear regression models incorporating each of the 36 significant lipid species, we also documented the estimated effect of maternal glycemia with the standardized birthweight and found that the corresponding covariates of the fasting and 2 h glucose measurements were both not associated with the standardized birthweight percentiles following FDR correction (Table 4). This suggests that maternal glycemia did not significantly influence the birthweight percentile, unlike the corresponding lipid species, in each model.

These results were corroborated when the birthweight outcome was considered categorically as SGA, AGA, and LGA. Even though our study was underpowered to show statistically significant associations following FDR correction, the topmost significantly different lipids between the groups still predominantly comprised lysophospholipids and TAGs, with changes going in the expected direction (Appendix A). At an uncorrected *p*-value < 0.05 compared with AGA, the SGA group showed 39 altered lipid concentrations (6 lysophospholipids were all lower, 16 TAGs were all higher, and 8 phospholipids and 9 ceramides were all higher) while the LGA group showed 16 altered lipid concentrations (7 lysophospholipids were all higher, 4 TAGs were all lower, 3 PEs, 1 PC, and 1 ceramide were all lower). Three of these lipids, LPC 20:2, TAG 54:1, and TAG 56:0, were found on both lists and were changed in the opposite direction when compared with the AGA group, consistent with results for the birthweight percentiles.

### 3.4. Sensitivity Analyses

In the first sensitivity analysis, which excluded the cases treated with insulin (Appendix A; *n* = 72), and in the second sensitivity analysis, which excluded the participants with diabetes in pregnancy (possibly undiagnosed pre-existing diabetes defined by fasting glucose ≥7.0 and/or 2 h glucose ≥11.1 mmol/L in a mid-gestation 75 g OGTT; Appendix A; *n* = 72), the list of lipid species most associated with the standardized birthweight percentiles, and the magnitudes of effect, were very similar to the list obtained in the main analyses of the complete set of subjects. The list still comprised only LPCs, LPEs, and TAGs with almost similar rankings even though there was overall less significant *p*-values, which are likely a reflection of the reduced statistical power of the smaller sample sizes. The third sensitivity analysis, which excluded the cases with possible placental insufficiency or hypertensive disorders of pregnancy (Appendix A), reduced the sample size considerably to 55, but still showed largely similar results. In all three sensitivity analyses, LPC 20:2 remained the topmost significantly associated with the standardized birthweight percentile, with a similar magnitude of association across all analyses (beta coefficient range 17.17–21.81% per-SD log_10_ lipid). Similar associations with birthweight percentiles were also obtained in unadjusted analyses of the whole sample, which included only the lipid species and the covariates of fasting and 2h maternal glucose (Appendix A) in the models.

## 4. Discussion

This study identified specific LPC and LPE species to be positively associated with birthweight, along with specific TAG species to be negatively associated with birthweight, independently of maternal glycemia. The cord plasma lipidome encapsulates maternal metabolism, placental metabolism, transfer across the maternal–placental–fetal axis and fetal metabolism, to provide valuable insights into the availability of nutrients for the fetus and the state of fetal metabolism as well as the presence of bioactive lipid signaling compounds circulating in the fetus [15,16]. The novel aspects of our study are the identification of the range of specific lipid species that could be potentially involved in the regulation of fetal size, and that individually they appear to have a greater influence on birthweight than maternal glycemia.

### 4.1. Cord Blood Lysophospholipids and Birthweight Percentile

Our results of the LPC lipid species strongly concur with a population-based observational study of predominantly White Caucasians in Germany [17] that also reported positive associations between cord blood LPC 16:1, LPC 18:1, and LPC 20:3 with birthweight. Other research has similarly indicated a strong and independent association between LPC 16:1 and birthweight [18]. Similarly, a UK study of pregnant women with obesity (UPBEAT cohort) also found positive correlations between cord blood LPC 16:1 or LPC 18:1 and birthweight using targeted metabolomics panels which tested only a restricted number of lipids. This suggests that such associations transcend ethnicity, implying that the potential role of LPC in fetal growth regulation is not substantially constrained by genetic, epigenetic, nutritional, and lifestyle factors. In addition, with our more extensive analytical method, we have made novel identifications of several other LPC species, including LPC 18:3, LPC 20:0, LPC 20:2, and LPC 22:5, that were positively associated with birthweight that have not previously been reported.

The high representation of LPCs containing mono-unsaturated fatty acids (MUFAs) such as LPC 16:1, LPC 18:1, LPC 20:1, and LPE 18:1, as among the most positively associated lipids with the birthweight percentiles is interesting. Stearoyl-Coenzyme A desaturase-1 (SCD1) is a rate-limiting enzyme that converts saturated fatty acids to mono-unsaturated fatty acids and is involved in the development of obesity-promoting lipogenesis [19]. Our results suggest that enhanced SCD1 activity, perhaps within the fetal–placental unit, may have a role in promoting fetal growth. Increased lipogenesis possibly results in greater fetal fat accretion and, consequently, a higher birthweight.

The main source of LPCs in the fetal circulation is unknown. A study has suggested the occurrence of the placental to fetal transfer of LPCs through plasma membrane transporters like MFSD2A [7] but endogenous fetal LPC synthesis is also likely. LPC as a lipid class is known to be related to the regulation of inflammatory processes [20,21] and LPC-DHA (docosahexaenoic acid) is important for brain growth [22] but their role in fetal metabolism, growth, and adiposity is not understood. Lysophospholipids comprise only a very small fraction of the total circulating fetal lipids; thus, they are unlikely to be significant contributors to the energy supply and cellular structures required for bodily growth, and are more likely to act as important biological signals regulating fetal growth. For example, LPCs were found to activate adipocyte glucose uptake, thereby lowering blood glucose in animal models of diabetes [23].

The most significant lipid species observed in every linear regression model, including all sensitivity analyses, was consistently LPC 20:2, which affirms the robustness of our results. The magnitude of the association of LPC 20:2 with birthweight is clinically significant; a twenty-one percentile increase in the standardized birthweight with every SD increase in log_10_-transformed concentration. The potential mechanism by which LPC 20:2 could regulate growth, however, remains elusive.

### 4.2. Cord Blood Triacylglycerols and Birthweight Percentile

Our cord lipidomic profiling showed that the TAG lipid species were negatively associated with the standardized birthweight percentiles. The measurement of the cord blood total TAGs in general using clinically validated instruments has been previously performed in several European studies [4,16,24], which also reported that the total TAGs were negatively associated with birthweight. They found that the TAG concentrations were significantly higher in small-for-gestational-age (SGA) neonates compared with appropriately grown and large-for-gestational-age (LGA) neonates. Unlike these studies, our study examined in greater depth what the specific TAG species were that were associated with birthweight, which is a novel contribution of our work. The fatty acid triplet composition of the 24 identified TAG species showed the predominance of the combinations of long-chain mono-unsaturated fatty acids with long-chain saturated fatty acids. Furthermore, about a fifth of these TAGs contained one odd chain fatty acid (C15 or C17) which are of very low abundance in the human circulation. The C17:0 fatty acid has been linked with glucose intolerance in adults, [25]; thus, its role as a metabolic modulator and growth regulator in the fetus is plausible.

We postulate that the negative correlation between the cord blood TAGs and birthweight may be partly attributed to lower endothelial lipoprotein lipase (LPL) activity in the adipose tissue of smaller fetuses, leading to the reduced hydrolysis of circulating TAG-rich lipoproteins [4,26,27], which is required to release fatty acids for local cellular uptake. The expression of local LPL is known to strongly associate with adipose tissue development [28]. Consistent with the notion of impaired lipid uptake and utilization by adipose tissue, Holtrop et al. [29] showed that TAGs were elevated in extremely low-birthweight infants receiving lipid emulsions. However, the fact that there are TAG species that are more negatively associated with birthweight than others suggest more specific functions played by individual TAGs over and above the supply of energy and substrates for growth. These roles remain speculative at present and may involve differential fatty-acid-composition-dependent substrate preferences of various tissue-specific lipases during development, which have implications for fetal growth.

### 4.3. Cord Blood Lipids and Maternal Glycemia

Within the regression models for each of the lipid species (lysophospholipids and TAGs) associated with birthweight, the influence of maternal glycemia on birthweight was limited. This is consistent with the postulation that fetal growth can also be significantly influenced by altered lipid supply and metabolism, and not just by maternal glycemia [5].

Nonetheless, we investigated if maternal glycemia was associated with any cord plasma lipid concentrations. The comparisons of our results with other studies that had used targeted metabolomics methods, which included a much narrower range of lipid species, showed some consistency. Like our study, one reported no association between cord lipids and maternal fasting glycemia [30], but unlike our results, another study found a positive association between maternal fasting glycemia and cord LPC 16:1 concentration [31]. Our finding of an association between the maternal 2 h post-glucose load and cord Cer d18:1/C33:1 is novel but needs to be replicated in an independent cohort. In general, ceramides are sphingolipids that form structural components of the plasma membrane as well as act as bioactive agents involved in regulating cell viability and oxygen sensing. They have been associated with changes in maternal insulin resistance peripartum [32] and in adults [33], and implicated in hepatic insulin resistance [34]. Either the altered transplacental passage or modified fetal synthesis of ceramides could plausibly underlie the ceramide association with maternal glycemia, but in our study, the ceramides did not associate with the birthweight percentiles and the clinical significance of this glycemia–ceramide association is unknown.

### 4.4. Limitations and Strengths of Study

Even though our findings are largely concordant with other similar studies, our study has several limitations that may reduce its generalizability. Our selected subsample only included women of Chinese ethnicity, who were predominantly non-obese. There were approximately uniform numbers from each glycemia quartile, which increases the representation at both ends of the glycemic spectrum and therefore does not represent normal distribution within a general obstetric population. Nevertheless, the GDM rate of 20% among this subsample is similar to the overall GDM incidence among Chinese women in the whole GUSTO cohort [35]. There was a higher proportion of SGA and LGA neonates, more than would be expected from a representative sample of the population, although these criteria were not used in the selection process. While the incidence of any hypertensive disorder (chronic and pregnancy-related) in this sub-study (12%) was somewhat higher than that in the overall GUSTO cohort (7.5%), it is nonetheless similar to the globally reported incidences [36]. However, the exclusion of the extreme cases of hyperglycemia, and of SGA and hypertension cases in the sensitivity analyses did not materially alter our results. We were unable to consider cord lipid associations with the OGTT 1 h post-load glucose, as this was not sampled since the two time-point WHO 1999 criteria was in use at the time of the study. Our expanded findings of lipid species associated with birthweight can largely be attributed to a more comprehensive analytical lipid profiling platform that covered over 400 lipid species distributed across 17 lipid classes. This allowed for the novel identification of specific lipid species that are independently and significantly associated with standardized birthweight percentiles. However, given our modest sample size and relatively narrow selection criteria, it is important to replicate our findings in a larger sample to confirm these associations.

## 5. Conclusions

In conclusion, our results demonstrate that birthweight is positively associated with specific fetal circulating lysophospholipids and negatively associated with specific TAGs, independently of maternal glycemia. We suggest that specific lysophospholipids may act as biological signals in promoting fetal growth, while the circulating concentrations of specific TAGs are changed because of altered metabolism and consumption by developing fetal tissues. Consistent with these postulations are our findings that lysophospholipids were the most prominent lipid class positively associated with LGA, possibly indicating their role in increased signaling to promote growth, while TAGs were the predominant lipid class associated with SGA, possibly reflecting reduced fetal utilization. Further studies to understand the mechanistic relationships between these specific lipid species and in utero growth could lead to new approaches in optimizing fetal size, including the modulation of transplacental lipid supply, which is urgently needed in an era of rising incidences of fetal macrosomia, maternal obesity, and diabetes.

## Figures and Tables

**Table 1 nutrients-16-00274-t001:** Characteristics of cases.

	Mean (SD) or	Range
	*n* (%)	(Lowest, Highest)
Maternal age (years)	31.91 (4.91)	(19.22, 41.10)
Pre-pregnancy BMI (kg/m^2^)	21.78 (4.01)	(15.30, 36.90)
Spontaneous conception	69 (92.0%)	
Parity		
Nulliparous	43 (57.3%)	
Multiparous	32 (42.7%)	
Plasma glucose in 75 g OGTT *^Ϯ^*		
Fasting glucose (mmol/L)	4.42 (0.62)	(3.60, 7.90)
2 h post-load glucose (mmol/L)	6.60 (1.65)	(3.30, 12.20)
Gestational diabetes ^§^	15 (20.0%)	
Hypertensive disorders		
Chronic hypertension Pre-eclampsia	1 (1.3%) 2 (2.7%)	
Pregnancy-induced hypertension	6 (8.0%)	
Female neonate	38 (50.7%)	
Gestational age at delivery (weeks)	39.15 (0.96)	(37.43, 41.14)
Mode of delivery		
Vaginal	51 (68.0%)	
Intrapartum cesarean section	10 (13.3%)	
Non-labor cesarean section	14 (18.7%)	
Standardized birthweight percentile ^#^	53.55 (34.79)	(1.25, 99.98)
Size at birth ^§^		
SGA (<10th percentile)	14 (18.7%)	
LGA (>90th percentile)	16 (21.3%)	

Data presented as mean (standard deviation) or number (%) unless otherwise stated. *^Ϯ^* Results from two time-point 75 g oral glucose tolerance test conducted at ~26 weeks’ gestation. ^§^ By WHO 1999 criteria (fasting glucose ≥ 7.0 and/or 2 h glucose ≥ 7.8 mmol/L). ^#^ Standardized for sex and gestational age using a local population reference calculated using methods described by Mikolajczyk et al., 2011 [13]. Abbreviations: BMI, body mass index; SGA, small-for-gestational-age; LGA, large-for-gestational-age.

**Table 2 nutrients-16-00274-t002:** Cord blood lipid species and their association with fasting glucose.

Lipid	Beta Coefficient (SD log_10_ Lipid per mmol/L Glucose) ^Ϯ^	LCL	UCL	*p*-Value	FDR- Corrected *p*-Value
DAG38:3	0.64	0.26	1.02	1.48 × 10^−3^	2.13 × 10^−1^
PC(O-36:2)	0.59	0.23	0.95	2.17 × 10^−3^	2.13 × 10^−1^
PC(P-36:1)	0.59	0.23	0.95	2.17 × 10^−3^	2.13 × 10^−1^
oddPC 35:2	0.56	0.2	0.93	3.49 × 10^−3^	2.13 × 10^−1^
DAG37:4	0.55	0.19	0.91	3.72 × 10^−3^	2.13 × 10^−1^
DAG38:2	0.55	0.19	0.92	3.99 × 10^−3^	2.13 × 10^−1^
SM d18:1/18:0	0.58	0.2	0.96	4.27 × 10^−3^	2.13 × 10^−1^
DAG43:4	0.56	0.18	0.93	4.83 × 10^−3^	2.13 × 10^−1^
PC(P-38:1)	0.58	0.19	0.97	4.93 × 10^−3^	2.13 × 10^−1^
DAG42:4	0.52	0.16	0.88	5.98 × 10^−3^	2.13 × 10^−1^
PC 32:2	0.55	0.17	0.93	6.16 × 10^−3^	2.13 × 10^−1^
DAG46:5	0.51	0.15	0.87	6.73 × 10^−3^	2.13 × 10^−1^
DAG37:3	0.53	0.15	0.91	7.37 × 10^−3^	2.13 × 10^−1^
SM d18:1/20:1	0.53	0.16	0.91	7.37 × 10^−3^	2.13 × 10^−1^
SM d18:1/18:1	0.53	0.15	0.91	8.15 × 10^−3^	2.19 × 10^−1^
PS 38:3	0.53	0.14	0.93	9.63 × 10^−3^	2.22 × 10^−1^
DAG39:1	0.5	0.12	0.88	1.19 × 10^−2^	2.22 × 10^−1^
DAG42:3	0.47	0.11	0.83	1.21 × 10^−2^	2.22 × 10^−1^
DAG43:5	0.48	0.11	0.85	1.42 × 10^−2^	2.22 × 10^−1^
MHCer 18:2/C22:0	0.48	0.11	0.86	1.46 × 10^−2^	2.22 × 10^−1^
MHCer 18:1/C20:0	0.51	0.11	0.91	1.47 × 10^−2^	2.22 × 10^−1^
DAG38:4	0.51	0.11	0.91	1.47 × 10^−2^	2.22 × 10^−1^
Cer d17:0/C23:1	0.51	0.11	0.91	1.48 × 10^−2^	2.22 × 10^−1^
Cer d18:1/C26:1	0.51	0.11	0.91	1.50 × 10^−2^	2.22 × 10^−1^
SM d18:1/22:0	0.48	0.1	0.86	1.53 × 10^−2^	2.22 × 10^−1^

^Ϯ^ Z-score standardized log_10_ lipid concentration per mmol/L increase in maternal fasting glucose concentration at ~26 weeks’ gestation. Adjusted for maternal age, maternal BMI, mode of delivery, assisted conception, and insulin treatment. FDR: false discovery rate; LCL, lower confidence limit; UCL, upper confidence limit.

**Table 3 nutrients-16-00274-t003:** Cord blood lipid species and their association with 2 h post-load glucose.

Lipid	Beta Coefficient (SD log_10_ Lipid per mmol/L Glucose) ^Ϯ^	LCL	UCL	*p*-Value	FDR- Corrected *p*-Value
Cer d18:1/C33:1	0.36	0.21	0.51	1.22 × 10^−5^	4.93 × 10^−3^
Cer d18:2/C16:0	0.3	0.14	0.45	3.27 × 10^−4^	6.61 × 10^−2^
Cer d19:0/C15:2	0.28	0.13	0.44	7.25 × 10^−4^	7.26 × 10^−2^
PC 36:3	0.26	0.11	0.41	1.24 × 10^−3^	7.26 × 10^−2^
PC(O-36:2)	0.25	0.1	0.39	1.48 × 10^−3^	7.26 × 10^−2^
PC(P-36:1)	0.25	0.1	0.39	1.48 × 10^−3^	7.26 × 10^−2^
LPC 14:0	0.25	0.1	0.4	1.57 × 10^−3^	7.26 × 10^−2^
DAG47:6	0.24	0.1	0.38	1.60 × 10^−3^	7.26 × 10^−2^
Cer d18:1/C33:0	0.27	0.11	0.43	1.76 × 10^−3^	7.26 × 10^−2^
LPC 16:1	0.24	0.1	0.39	1.80 × 10^−3^	7.26 × 10^−2^
PC(P-38:1)	0.25	0.1	0.41	2.17 × 10^−3^	7.96 × 10^−2^
LPC 20:2	0.23	0.09	0.37	2.45 × 10^−3^	8.07 × 10^−2^
PE(P-18:0/22:5)	0.24	0.09	0.39	2.71 × 10^−3^	8.07 × 10^−2^
DAG43:4	0.24	0.09	0.39	2.80 × 10^−3^	8.07 × 10^−2^
oddPC 35:2	0.22	0.07	0.37	4.54 × 10^−3^	1.22 × 10^−1^
PC 38:3	0.23	0.07	0.38	5.33 × 10^−3^	1.35 × 10^−1^
LPC 16:0	0.22	0.07	0.38	6.20 × 10^−3^	1.44 × 10^−1^
PE(O-18:2/20:3)	0.22	0.07	0.37	6.44 × 10^−3^	1.44 × 10^−1^
PE(O-40:6)	0.21	0.06	0.37	7.76 × 10^−3^	1.51 × 10^−1^
PE(P-16:0/20:4)	0.22	0.06	0.38	8.21 × 10^−3^	1.51 × 10^−1^
PE(P-36:4)	0.22	0.06	0.38	8.21 × 10^−3^	1.51 × 10^−1^
Cer d18:1/C26:0	0.23	0.06	0.39	8.23 × 10^−3^	1.51 × 10^−1^
DAG46:6	0.2	0.05	0.35	9.16 × 10^−3^	1.61 × 10^−1^
DAG47:5	0.2	0.05	0.34	1.01 × 10^−2^	1.71 × 10^−1^
Cer d18:1/C28:0	0.21	0.05	0.37	1.16 × 10^−2^	1.78 × 10^−1^

^Ϯ^ Z-score standardized log_10_ lipid concentration per mmol/L increase in 2 h maternal glucose concentration in a 75 g oral glucose tolerance test conducted at ~26 weeks’ gestation. Adjusted for maternal age, maternal BMI, mode of delivery, assisted conception, and insulin treatment. Statistical significance set at *p* < 0.05. FDR: false discovery rate; LCL, lower confidence limit; UCL, upper confidence limit.

**Table 4 nutrients-16-00274-t004:** Cord blood lipids associated with birthweight percentile and the relative influence of the covariates of maternal fasting and 2 h post-load glucose.

	Association between Lipid and Standardized Birthweight Percentile ^†^				Fasting Glucose Influence on Standardized Birthweight Percentile ^†^				2 h Glucose Influence on Standardized Birthweight Percentile ^†^			
Lipid Species (Predictor)	Beta Coefficient (BW%/SD log_10_ Lipid) ^††^	Lipid LCL	Lipid UCL	FDR- Corrected *p*-Value	Beta Coefficient (BW%/mmol/L) ^†††^	Lipid LCL	Lipid UCL	FDR- Corrected *p*-Value	Beta Coefficient (BW%/mmol/L) ^†††^	Lipid LCL	Lipid UCL	FDR- Corrected *p*-Value
LPC 20:2	21.28	12.7	29.87	2.25 × 10^−3^	11.19	−6.87	29.25	4.51 × 10^−1^	−3.86	−11.42	3.7	9.79 × 10^−1^
LPC 18:1	18.99	11	26.99	2.25 × 10^−3^	9.55	−8.71	27.81	4.51 × 10^−1^	−1.84	−9.33	5.64	9.79 × 10^−1^
LPC 16:1	18.38	9.38	27.38	7.07 × 10^−3^	11.95	−6.96	30.85	4.51 × 10^−1^	−3.66	−11.62	4.31	9.79 × 10^−1^
LPC 18:2	16.42	8.29	24.55	7.07 × 10^−3^	11.38	−7.56	30.32	4.51 × 10^−1^	−2.49	−10.34	5.36	9.79 × 10^−1^
LPE 18:1	16.15	8.15	24.15	7.07 × 10^−3^	4.34	−14.85	23.54	6.61 × 10^−1^	3.18	−4.58	10.94	9.79 × 10^−1^
LPC 20:3	15.87	7.14	24.6	1.51 × 10^−2^	10.32	−8.98	29.62	4.51 × 10^−1^	−1.94	−9.9	6.03	9.79 × 10^−1^
LPC 18:3	15.61	6.32	24.9	2.43 × 10^−2^	17.93	−2.08	37.93	4.51 × 10^−1^	−4	−12.43	4.42	9.79 × 10^−1^
LPC 22:5	14.71	6.14	23.27	2.23 × 10^−2^	10.66	−8.81	30.14	4.51 × 10^−1^	−1.24	−9.23	6.74	9.79 × 10^−1^
LPC 20:1	13.35	5.03	21.67	3.28 × 10^−2^	5.52	−14.4	25.44	5.92 × 10^−1^	0.39	−7.58	8.36	9.79 × 10^−1^
LPE 18:2	13.35	5.3	21.4	2.66 × 10^−2^	7.78	−11.87	27.43	4.64 × 10^−1^	0.41	−7.53	8.34	9.79 × 10^−1^
LPC 20:0p/20:1e	13.32	5.49	21.15	2.31 × 10^−2^	12.88	−6.67	32.42	4.51 × 10^−1^	−1.58	−9.61	6.45	9.79 × 10^−1^
LPC 20:4	13.19	4.53	21.85	4.61 × 10^−2^	12.45	−7.38	32.27	4.51 × 10^−1^	−1.37	−9.52	6.78	9.79 × 10^−1^
TAG50:2	−27.67	−41.81	−13.52	8.57 × 10^−3^	9.16	−9.91	28.22	4.51 × 10^−1^	−0.05	−7.78	7.68	9.97 × 10^−1^
TAG50:4	−25.47	−36.7	−14.24	2.39 × 10^−3^	10.39	−8.08	28.85	4.51 × 10^−1^	0.67	−6.81	8.15	9.79 × 10^−1^
TAG54:3	−20.27	−31.31	−9.24	1.47 × 10^−2^	12.73	−6.57	32.02	4.51 × 10^−1^	−0.2	−8.02	7.63	9.87 × 10^−1^
TAG50:3	−19.96	−30.18	−9.74	8.57 × 10^−3^	11.29	−7.77	30.35	4.51 × 10^−1^	0.01	−7.73	7.74	9.99 × 10^−1^
TAG54:1	−19.93	−29.07	−10.79	3.76 × 10^−3^	13.77	−4.91	32.46	4.51 × 10^−1^	−1.04	−8.64	6.56	9.79 × 10^−1^
TAG52:3	−18.81	−27	−10.61	2.38 × 10^−3^	13.8	−4.66	32.27	4.51 × 10^−1^	−0.51	−7.99	6.97	9.79 × 10^−1^
TAG58:1	−18.6	−28.21	−8.99	8.93 × 10^−3^	8.91	−10.2	28.02	4.51 × 10^−1^	0.54	−7.2	8.28	9.79 × 10^−1^
TAG52:2	−18.48	−26.14	−10.83	2.25 × 10^−3^	10.66	−7.52	28.85	4.51 × 10^−1^	−0.64	−8.03	6.76	9.79 × 10^−1^
TAG56:0	−17.37	−24.81	−9.93	2.28 × 10^−3^	12.84	−5.52	31.21	4.51 × 10^−1^	−0.02	−7.46	7.42	9.98 × 10^−1^
TAG54:2	−16.76	−24.76	−8.76	5.90 × 10^−3^	14.93	−3.98	33.83	4.51 × 10^−1^	−1.2	−8.87	6.48	9.79 × 10^−1^
TAG58:2	−16.31	−25.29	−7.33	1.51 × 10^−2^	9	−10.32	28.33	4.51 × 10^−1^	0.79	−7.03	8.62	9.79 × 10^−1^
TAG52:1	−16.31	−24.87	−7.74	1.03 × 10^−2^	11.94	−7.21	31.1	4.51 × 10^−1^	−0.8	−8.6	7.01	9.79 × 10^−1^
TAG52:4	−16.03	−24.22	−7.85	8.57 × 10^−3^	12.01	−7.04	31.07	4.51 × 10^−1^	0.33	−7.39	8.05	9.80 × 10^−1^
TAG53:0	−16.02	−25.53	−6.5	2.43 × 10^−2^	10.61	−8.92	30.14	4.51 × 10^−1^	−0.67	−8.63	7.3	9.79 × 10^−1^
TAG54:0	−14.87	−23.12	−6.62	1.55 × 10^−2^	8.63	−10.73	27.99	4.51 × 10^−1^	0.66	−7.17	8.49	9.79 × 10^−1^
TAG53:1	−14.76	−23.34	−6.19	2.23 × 10^−2^	10.41	−9.06	29.87	4.51 × 10^−1^	0.17	−7.73	8.07	9.87 × 10^−1^
TAG55:3	−14.73	−23.32	−6.14	2.23 × 10^−2^	7.46	−12.1	27.03	4.73 × 10^−1^	0.99	−6.9	8.89	9.79 × 10^−1^
TAG57:3	−14.66	−23.71	−5.61	3.10 × 10^−2^	9.83	−9.81	29.48	4.51 × 10^−1^	0.17	−7.79	8.14	9.87 × 10^−1^
TAG53:3	−14.3	−23.07	−5.54	3.00 × 10^−2^	8.02	−11.67	27.7	4.63 × 10^−1^	0.77	−7.18	8.71	9.79 × 10^−1^
TAG48:3	−14.27	−22.59	−5.94	2.23 × 10^−2^	11.15	−8.33	30.63	4.51 × 10^−1^	0.82	−7.08	8.71	9.79 × 10^−1^
TAG48:2	−13.24	−21.86	−4.62	4.61 × 10^−2^	9.99	−9.79	29.77	4.51 × 10^−1^	0.9	−7.11	8.91	9.79 × 10^−1^
TAG52:0	−12.62	−20.93	−4.32	4.61 × 10^−2^	7.93	−11.95	27.81	4.64 × 10^−1^	0.92	−7.1	8.94	9.79 × 10^−1^
TAG58:3	−12.4	−20.57	−4.24	4.61 × 10^−2^	8.18	−11.69	28.05	4.62 × 10^−1^	0.58	−7.45	8.6	9.79 × 10^−1^
TAG56:1	−12.11	−20.08	−4.15	4.61 × 10^−2^	8.47	−11.38	28.32	4.53 × 10^−1^	0.56	−7.46	8.58	9.79 × 10^−1^

^†^ Standardized for sex and gestational age using a local population reference calculated using methods described by Mikolajczyk et al. 2011. [13]. ^††^ Standardized birthweight percentile change for each SD increase in z-score standardized log_10_ lipid concentration. ^†††^ Standardized birthweight percentile change per mmol/L increase in maternal glucose concentration (fasting and 2 h) determined in a 75 g oral glucose tolerance test conducted at ~26 weeks’ gestation. Regression model: BW percentile as outcome, cord blood lipid concentration as predictor, adjusted for these covariates: maternal age, maternal BMI, parity, mode of delivery, assisted conception, fasting glucose, 2 h post-load glucose, insulin treatment. Statistical significance set at *p* < 5.00 × 10^−2^. Abbreviations: BW%, standardized birthweight percentile; FDR, false discovery rate; LCL, lower confidence limit; UCL, upper confidence limit.

## Data Availability

Data are available upon reasonable request from the GUSTO team.

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
