# Peer review of "Umbilical Cord Plasma Lysophospholipids and Triacylglycerols Associated with Birthweight Percentiles"

_nutrients, 2024, doi:10.3390/nu16020274_

Round 1
Reviewer 1 Report
Comments and Suggestions for Authors
Author Response
We thank the reviewer for his/her suggestions and comments which we have considered thoroughly.
- Please improve the inclusion and exclusion criteria, and finally include 75 pregnant women by which inclusion and exclusion criteria. The first paragraph of methodology is the inclusion and exclusion criteria of the whole GUSTO design.
We have extensively revised our description of the sample selection process and edited out the less relevant details of the eligibility criteria for the whole GUSTO cohort. We have now also included a Supplementary Figure S1 illustrating the study selection flow chart.
Lines 99-116:
“Pregnant women (n=1450) were recruited at 10 to 14 weeks of gestation between June 2009 and September 2010 from the two largest public maternity units in Singapore, KKH and NUH. Eligible participants were Singapore residents of Chinese, Malay or Indian ethnicity, willing to donate bio-samples. Further details of the overall GUSTO study have been described elsewhere [9]. The majority of participants completed a two time-point 75g oral glucose tolerance test (OGTT; n=1136) and provided cord blood samples (n=886).
For this sub-study, cases were selected based on availability of cord plasma samples and data on OGTT, pre-pregnancy BMI and ethnicity. Known cases of pre-existing maternal diabetes, multiple gestation and preterm delivery, were excluded. To facilitate further selection, both the fasting and 2h glycemia values were each considered in quartiles (as calculated across the whole cohort); cases where both the fasting and 2h glycemia lied in the same quartile were considered for selection. Cases were then also confined to those of neonates with homogenous Chinese parentage (the largest ethnic group in GUSTO) [9] and delivered by non-smoking mothers, to minimize confounding by these factors. This resulted in a final sample of 75 cases (study selection flow chart in Supplementary Figure S1), which incidentally comprised a similar number of cases in each of the four glycemia quartiles: 19, 21, 17 and 18 cases in quartiles one to four, respectively.”
- Whether 75 pregnant women were statistically significant.
We have now acknowledged more explicitly the limitation of our sample size in the discussion and have edited lines 406-407 to state that “given our modest sample size and relatively narrow selection criteria it is important to replicate our findings in a larger sample to confirm these associations.”
- 75g 2h-OGTT, 1h-OGTT were not considered
At the time of the GUSTO study (2009-2010), the WHO 1999 criteria for diagnosing GDM using the two time point OGTT was the one used in clinical practice, as we have stated on lines 160-162. The 1h glucose sample was therefore not collected. We have now added this as a limitation on lines 400-402. “We were unable to consider cord lipid associations with the OGTT 1h post-load glucose as this was not sampled since the two-time-point WHO 1999 criteria was in use at the time of the study.”
- Is there a LOD lower limit for lipid metabolome detection, and if so, how to process the data?
Signals falling below the limit of quantitation (LOQ) were reported as absent/not found (i.e. assigned a value of 0). We have now added a line in the Methods stating so. Line 137 “Signals falling below the limit of quantitation (LOQ) were assigned a value of 0.”
- Descriptive analysis of the results of lipid detection is recommended, including Min, 25%, 50%, 75% and max for presentation.
Raw data of peak area values were normalised with the peak area of the corresponding internal standard to give the normalized concentration of the lipid species in μg/mL. The Min, 25%, 50%, 75% and max concentration for lipid species are now presented in Supplementary Table S1. We have now added further detail in the Methods on lines 135-141,
“Samples were analysed in a single batch where quality control (QC) samples, prepared by pooling lipid extracts from all samples, were inserted after every 10 samples.”
“In total, the concentrations of 404 lipid species from 17 lipid classes were determined; peak area for each species was normalized against respective internal standards. Concentrations of these lipids are detailed in Supplementary Table S1.”
We have also added an additional reference describing the validation method in more detail.
- It is recommended to supplement the analysis of LGA/SGA/macrosomia/LBW
We have performed new analyses associating lipids with the mutually exclusive categories of SGA, AGA, LGA, describing this in the Methods section (lines 162-166) and summarised the results on lines 258-269.
“Logistic regression adjusting for the same covariates were additionally performed to associate lipids with the mutually exclusive categories of small-for-gestational-age (SGA; birthweight <10th percentile) and large-for-gestational-age (LGA; >90th percentile), each group compared with the appropriately-grown-for-gestational age (AGA; 10-90th percentile) group.”
“These results were corroborated when birthweight outcome was considered categorically as SGA, AGA and LGA. Even, though our study was underpowered to show statis-tically significant associations following FDR-correction, the topmost significantly differ-ent lipids between groups still predominantly comprised lysophospholipids and TAGs, with changes going in the expected direction (Supplementary Table S2). At an uncorrected p-value<0.05 compared with AGA, the SGA group showed 39 altered lipid concentrations (6 lysophospholipids were all lower, 16 TAGs were all higher, and 8 phospholipids and 9 ceramides were all higher) while the LGA group showed 16 altered lipid concentrations (7 lysophospholipids were all higher, 4 TAGs were all lower, 3 PEs, 1 PC and 1 ceramide were all lower). Three of these lipids, LPC 20:2, TAG 54:1, and TAG 56:0, were found on both lists and were changed in the opposite direction when compared with the AGA group, consistent with results for birthweight percentiles.
Data is also included in a new table as supplementary data (Supplementary Table S2)
We have also discussed these results in Discussion lines 415-418
“Consistent with these postulations are our findings that lysophospholipids were the most prominent lipid class positively associated with LGA possibly indicating their role in increased signaling to promote growth, while TAGs were the predominant lipid class associated with SGA possibly reflecting reduced fetal utilization.”
- What is the incidence of GDM/hypertensive disorders of pregnancy in GUSTO, and whether the small number of GDM and pregnancy-induced hypertension in these 75 samples is representative.
As mentioned on line 391, the GDM incidence in our sub-study of 20% is similar to the overall GDM incidence in the GUSTO cohort of 18.9% (Chong et.al. BMC Pregnancy and Childbirth 345 (2014), and also similar to that among the Chinese alone which was 21%. While the incidence of any hypertensive disorder in this sub-study was 12%, which is somewhat higher than the rate in the overall GUSTO cohort of 7.5%, it is similar to the globally reported incidences of between 4-25% (Wang et.al. BMC Pregnancy Childbirth 2021, 21, 364). We have now added to lines 395-398: “While the incidence of any hypertensive disorder (chronic and pregnancy-related) in this sub-study (12%) was somewhat higher than that in the overall GUSTO cohort (7.5%), it is nonetheless similar to the globally reported incidences.”
- As a large part of the BMI data of mothers involved in the article came from their own descriptions, will the error have a certain impact on the evaluation of the results.
We have previously reported that the pre-pregnancy BMI based on self-reported weight was highly correlated with 1st trimester BMI calculated using measured weight and height at the first antenatal visit (correlation coefficient=0.96, p<0.001) (Loy et.al. BMC Public Health 2019). We have added this detail to lines 119-120.
“….self-reported pre-pregnancy body mass index (BMI; which was highly correlated with measured first trimester BMI in the overall GUSTO cohort, correlation coefficient 0.96, p<0.001)”
- The design of the article Table 1 is not very clear and beautiful. You can add another column to place the refined category after the larger category, for example.
The format of this table was disrupted during article processing. We have now corrected it back to the format it was meant to be in.
Reviewer 2 Report
Comments and Suggestions for Authors
Thanks to the authors for submitting their interesting manuscript entitled “Umbilical Cord Plasma Lysophospholipids and Triacylglycerols Associated with Birthweight Percentiles”. I have carefully reviewed the manuscript. This is very interesting research. I want to provide them with feedback and recommendations that could enhance the manuscript’s clarity and impact.
Please find my comments below.
· Line 102: The information about eligible participants being of Chinese, Malay, or Indian ethnicity, Singapore residents with the intention to reside in Singapore for at least five years post-delivery, and willing to donate bio-samples might require clarification regarding its relevance to the study, especially considering that the results primarily focus on cord blood values.
· Line 104: The exclusion criteria for women receiving chemotherapy, psychotropic drugs, or those with type 1 diabetes mellitus is intriguing. Including an explanation for these specific groups’ exclusions would provide better context and understanding to the readers.
· Line 277: The statement regarding associations transcending ethnicity is intriguing. It might be beneficial to expand on the term “ethnicity,” touching upon aspects like genetics, epigenetics, nutritional habits, and lifestyle factors that could influence these associations.
· Line 312: Considering that the cord blood values reflect the intrauterine environment, expanding the discussion to include infants classified as IUGR (Intrauterine Growth Restriction) alongside SGA babies born below the 10th percentile could provide a more comprehensive understanding of the intrauterine environment and outcomes.
I hope these suggestions contribute to refining the manuscript further. Overall, the study is impressive and contributes significantly to the field.
Comments on the Quality of English LanguageMinor editing of English language required
Author Response
We thank the reviewer for his/her encouraging and constructive comments.
Line 102: The information about eligible participants being of Chinese, Malay, or Indian ethnicity, Singapore residents with the intention to reside in Singapore for at least five years post-delivery, and willing to donate bio-samples might require clarification regarding its relevance to the study, especially considering that the results primarily focus on cord blood values.
Line 104: The exclusion criteria for women receiving chemotherapy, psychotropic drugs, or those with type 1 diabetes mellitus is intriguing. Including an explanation for these specific groups’ exclusions would provide better context and understanding to the readers.
In response to comments by the other reviewer as well as this comment, the selection criteria have now been streamlined, and additional relevant information is provided related to this sub-study. The eligibility criteria for the overall GUSTO cohort have now been curtailed and only the essential information is included. A reference is included for interested readers to find out more about the overall study eligibility criteria. We have now also included a Supplementary Figure S1 illustrating the study selection flow chart.
Lines 99-116:
“Pregnant women (n=1450) were recruited at 10 to 14 weeks of gestation between June 2009 and September 2010 from the two largest public maternity units in Singapore, KKH and NUH. Eligible participants were Singapore residents of Chinese, Malay or Indian ethnicity, willing to donate bio-samples. Further details of the overall GUSTO study have been described elsewhere [9]. The majority of participants completed a two time-point 75g oral glucose tolerance test (OGTT; n=1136) and provided cord blood samples (n=886).
For this sub-study, cases were selected based on availability of cord plasma samples and data on OGTT, pre-pregnancy BMI and ethnicity. Known cases of pre-existing maternal diabetes, multiple gestation and preterm delivery, were excluded. To facilitate further selection, both the fasting and 2h glycemia values were each considered in quartiles (as calculated across the whole cohort); cases where both the fasting and 2h glycemia lied in the same quartile were considered for selection. Cases were then also confined to those of neonates with homogenous Chinese parentage (the largest ethnic group in GUSTO) [9] and delivered by non-smoking mothers, to minimize confounding by these factors. This resulted in a final sample of 75 cases (study selection flow chart in Supplementary Figure S1), which incidentally comprised a similar number of cases in each of the four glycemia quartiles: 19, 21, 17 and 18 cases in quartiles one to four, respectively.”
Line 277: The statement regarding associations transcending ethnicity is intriguing. It might be beneficial to expand on the term “ethnicity,” touching upon aspects like genetics, epigenetics, nutritional habits, and lifestyle factors that could influence these associations.
We have now elaborated on this sentence in lines 304-306: “This suggests that such associations transcend ethnicity, implying that the potential role of LPC in fetal growth regulation is not substantially constrained by genetic, epigenetic, nutritional and lifestyle factors.”
Line 312: Considering that the cord blood values reflect the intrauterine environment, expanding the discussion to include infants classified as IUGR (Intrauterine Growth Restriction) alongside SGA babies born below the 10th percentile could provide a more comprehensive understanding of the intrauterine environment and outcomes.
In response to Reviewer 1 comments, we have already performed additional analyses for the categorical outcomes of SGA and LGA compared with AGA. Only 6 out of the 14 SGA cases demonstrated fetal growth deceleration (ultrasound-measured 2nd-3rd-trimester abdominal circumference decrease by ≥0.67 standard deviation score, as we have previously defined; J Hypertens. 2022 Nov 1;40(11):2171-2179) representing likely intrauterine growth restriction. We were therefore underpowered to perform further analyses of subgroups of the SGA cases.